# A symbiotic physical niche in *Drosophila melanogaster* regulates stable association of a multi-species gut microbiota

Ren Dodge[1], Eric W. Jones[2,3], Haolong Zhu [1,4], Benjamin Obadia[5], Daniel J. Martinez[1], Chenhui Wang[1,6], Andrés Aranda-Díaz[7], Kevin Aumiller[1,4], Zhexian Liu[4], Marco Voltolini [8,9], Eoin L. Brodie [8], Kerwyn Casey Huang [7,10,11], Jean M. Carlson[3], David A. Sivak [2], Allan C. Spradling[1,4,6] & William B. Ludington [1,4] ✉

The gut is continuously invaded by diverse bacteria from the diet and the environment, yet microbiome composition is relatively stable over time for host species ranging from mammals to insects, suggesting host-specific factors may selectively maintain key species of bacteria. To investigate host specificity, we used gnotobiotic *Drosophila*, microbial pulse-chase protocols, and microscopy to investigate the stability of different strains of bacteria in the fly gut. We show that a host-constructed physical niche in the foregut selectively binds bacteria with strain-level specificity, stabilizing their colonization. Primary colonizers saturate the niche and exclude secondary colonizers of the same strain, but initial colonization by *Lactobacillus* species physically remodels the niche through production of a glycan-rich secretion to favor secondary colonization by unrelated commensals in the *Acetobacter* genus. Our results provide a mechanistic framework for understanding the establishment and stability of a multi-species intestinal microbiome.

Host health is affected by the composition of the gut microbiome, specifically which species and strains of bacteria occupy the gut[1–5]. The microbiome is established and maintained in the face of daily fluctuations in diet, invasion by pathogens[6], and disruptions by antibiotics[7]. Many gut resident bacteria localize to specific regions of the gut that correspond to chemical environments matching the specific species's metabolism[8]. Certain probiotics, namely *Lactobacillus* species, additionally make physical attachments with host mucus, stabilizing their colonization[9,10]. With strain-level diversity in the hundreds to thousands[11], it remains enigmatic how a host can select and maintain a specific set of strains. One hypothesis is that long-term

maintenance of diet and lifestyle habits reinforce microbiome stability[12–16], while another, non-exclusive hypothesis is that the host constructs physical niches in the gut that acquire and sequester specific symbiotic bacteria[17–22].

The microbiome of the fruit fly, *Drosophila melanogaster*, has been studied for over a century and is relatively simple in its composition compared to the mammalian gut microbiome[23], yet how fly gut microbiome assembly is regulated remains unclear. Similar to mammalian colonic crypts, the fly gut is microaerobic and colonized by bacteria from the Lactobacillales class and Proteobacteria phylum[22,24–26]. Flies can easily be reared germ-free and then associated with defined

[1]Department of Embryology, Carnegie Institution for Science, Baltimore, MD 21218, USA. [2]Department of Physics, Simon Fraser University, Burnaby, BC V5A 1S6, Canada. [3]Department of Physics, University of California, Santa Barbara, CA 93106, USA. [4]Department of Biology, Johns Hopkins University, Baltimore, MD 21218, USA. [5]Molecular and Cell Biology Department, University of California, Berkeley, CA 94720, USA. [6]Howard Hughes Medical Institute, Baltimore, MD 21218, USA. [7]Department of Bioengineering, Stanford University, Stanford, CA 94305, USA. [8]Lawrence Berkeley National Lab, Berkeley, CA 94720, USA. [9]Dipartimento di Scienze della Terra, Università degli Studi di Milano, Milano, Italy. [10]Department of Microbiology and Immunology, Stanford University School of Medicine, Stanford, CA 94305, USA. [11]Chan Zuckerberg Biohub, San Francisco, CA 94158, USA. ✉e-mail: ludington@carnegiescience.edu

bacterial strains, providing a high level of biological control[27]. Furthermore, the fly gut microbiome has low diversity, with ~5 species of stable colonizers from two primary groups: the genera *Lactobacillus* (phylum Firmicutes), which was recently split into *Lactiplantibacillus* and *Levilactibacillus*, and *Acetobacter* (class α-Proteobacteria)[26,28]. These species are easily cultured, genetically tractable[27], and they affect fly lifespan, fecundity, and development[29–35]. While colonization of the fly gut has long been argued to be non-specifically regulated by host filtering mechanisms, including feeding preferences, immunity, and digestion, recent evidence suggests flies may also selectively acquire *Lactobacillus* and *Acetobacter* strains in the wild[24,36], and these may provide flies with nutrition during the larval phase[37].

Here, we discover a physical niche within the adult *Drosophila* foregut that is specifically colonized by wild strains of *Lactobacillus* and *Acetobacter*. We characterize the spatial specificity of the niche, the bacterial strain specificity for colonization, and the stability of colonization. We measure priority effects that regulate the order in which the bacterial species colonize. Finally, we measure the response of the niche to bacterial colonizers including physical changes and glycosylation of the extracellular matrix.

## Results

### Spatially specific gut localization of *Lactiplantibacillus plantarum* from wild flies

In previous work, we surveyed a range of bacterial strains that were associated with either lab or wild-caught *D. melanogaster*[24], identifying a subset of bacterial strains that efficiently colonize the gut of lab flies. To investigate whether commensal bacteria form stable associations with the fly gut in a manner consistent with the existence of a niche, we exposed flies to a quantified inoculum of bacterial cells labeled with a fluorescent protein (Fig. S1A–G). Following inoculation, flies were transferred to germ-free food daily for 3 d followed by an additional transfer to a new germ-free vial for 3 h to allow transient bacteria to clear from the gut ("Methods", Fig. S1). Clearing prior to analysis reduced the total number of gut bacteria and the spatial variation in bacterial location (Fig. S1H–J). These experiments revealed that a strain of *Lactiplantibacillus plantarum* (*Lp*) isolated from a wild-caught fly (*LpWF*) persists exclusively in the *D. melanogaster* foregut (Figs. 1A–D, S1I,J), including the proventriculus (a luminal region connecting the esophagus with the anterior midgut[38]), the crop (a sack-like appendage), and the crop duct that connects the crop to the proventriculus. Bacteria associated with longitudinal furrows lining the surface of the proventriculus inner lumen, the crop duct, and the base of the crop (Fig. 1C-D, S1J). Similar spatial patterns of colonization occurred in adult mated females, virgin females, and adult males (Fig. S2A–J), indicating that sex-specific differences do not affect the phenotype. No colonization was observed in larvae post-clearing, indicating that the phenotype is specific to the adult foregut (Fig. S2K–M). We focus on adult mated females throughout because they have been shown to exhibit low fly-to-fly variation in gut phenotypes[39–41].

Similar to *LpWF*, a strain of *Acetobacter indonesiensis* colonized the same foregut regions (Figs. 1E, S2, S3), indicating that the two major groups of fly gut bacteria have the same spatial specificity in the foregut. By contrast, flies colonized with *Lp* from laboratory flies (*LpLF*) (Fig. S1K, L) or the *LpWCFS1* strain isolated from humans (Figs. S1M, S2A–J) exhibited much lower levels of colonization. No *Lp* strains were found at substantial abundance in the midgut or other regions of the fly after clearing transient bacteria. Consistent with microscopy, live bacterial density was greatest in the proventriculus, followed by the crop, and was lowest in the midgut and hindgut (Figs. 1F, S1N). We further validated that *LpWF* maintains stable colonization in the absence of ingestion of new bacterial cells over 5 d during which non-adherent bacteria were flushed from the gut by fastidiously maintaining sterility of the food using a CAFÉ feeder[24] (Fig. S4A, B).

A bacterial population in the foregut with the observed spatial localization might be maintained by proliferation and constant re-seeding from the crop, in which case flies without crops could not be stably colonized. We conducted microsurgery to remove the crop from germ-free flies (Fig. 1G, "Methods"), inoculated them with *LpWF* 5 d post-surgery, and then dissected and imaged the gut 5 d post inoculation (dpi). Surgical success was validated and the remaining portion of the crop duct had a melanized scar at the surgery site (Fig. 1H). All cropless flies were stably colonized by *LpWF* ($n = 15/15$), with a high density of bacteria in the proventriculus inner lumen as in flies with an intact crop (Fig. 1C). We observed similar *Ai* colonization following cropectomy ($n = 14/14$ colonized, Fig. S3E, F). Examining these regions further, transmission electron microscopy (TEM) of the proventriculus lumen revealed a consistent tissue geometry (Fig. 1I, J), with densely packed bacterial cells longitudinally oriented in elongated furrows formed by host cell bodies making up an average of 11 ridges per cross section (Fig. S4C).

Thus, the crop is not required for stable foregut colonization by *LpWF* or *Ai*, suggesting that the ability to specifically bind to the proventriculus and crop duct is key to stable bacterial association.

### Commensal association saturates at a precise bacterial population size and resists displacement, consistent with a niche

A niche would be expected to result in strong bacterial association based on specific binding sites, such that the associated bacterial population size would saturate at a well-defined value. Moreover, cells already bound to the proventriculus would be expected to promote population stability and prevent later-arriving bacteria from colonizing. To test these hypotheses, we colonized germ-free flies with a range of doses of *LpWF*-mCherry and measured the abundance over time. As predicted, over a wide range of initial inoculum sizes, the associated bacterial population saturated at ~$10^4$ CFUs/fly (Fig. 2A). Furthermore, when the inoculum size was below that saturation level, the population of bacteria in the proventriculus increased gradually and plateaued within 5 d. Growth measurements in live flies[24] demonstrated that the plateau was reached by growth of the initially bound population rather than ingestion of additional cells. By contrast, when an excess of bacteria was supplied initially, the population decreased to the same plateau value within 1 d (Fig. 2A), indicating that the niche has a finite and fixed carrying capacity. Similar dynamics were observed for *Ai* with ~$10^3$ cells at the saturated density (Fig. S3G).

To investigate the stability of bacterial colonization in the proventriculus, we performed a pulse-chase experiment in which we challenged *LpWF*-mCherry-pre-colonized flies with unlabeled *LpWF* fed in excess over the course of 10 d (Fig. 2B). *LpWF*-mCherry levels in the gut decreased by >90% over the first 5 d, from ~$10^4$ to ~$10^3$ CFUs/fly, and then remained at ~$10^3$ CFUs/fly for the following 5 d (Fig. 2C), indicating a small, bound population with little turnover and a larger associated population with a half-life of 2.5 d (95% confidence interval (c.i.) 1.6–4.3 d). By contrast, *LpWCFS1*, a weakly-colonizing human isolate of *L. plantarum*, was quickly flushed from the gut (Fig. 2C). Similar dynamics were observed in *Ai* (Fig. 2C, S3H) with a half-life of 2.5 d (95% c.i. 1.3–6.5 d), indicating that the niche has equivalent kinetics for both bacterial species.

Initial binding to the niche is a key step in the establishment of a new bacterial population prior to filling the niche. Establishment is dose-dependent[24], and our finding that the final abundance of late colonizers is lower than that of initial colonizers (Fig. S4D) suggested that the presence of prior colonizers would shift the dose-response curve. To quantify such priority effects, we fed a range of doses of *LpWF*-mCherry to individual *LpWF*-pre-colonized flies and measured the percentage that were colonized by *LpWF*-mCherry 3 d later. Consistent with our hypothesis, pre-colonized flies were less likely than germ-free flies to become colonized by an equal dose of

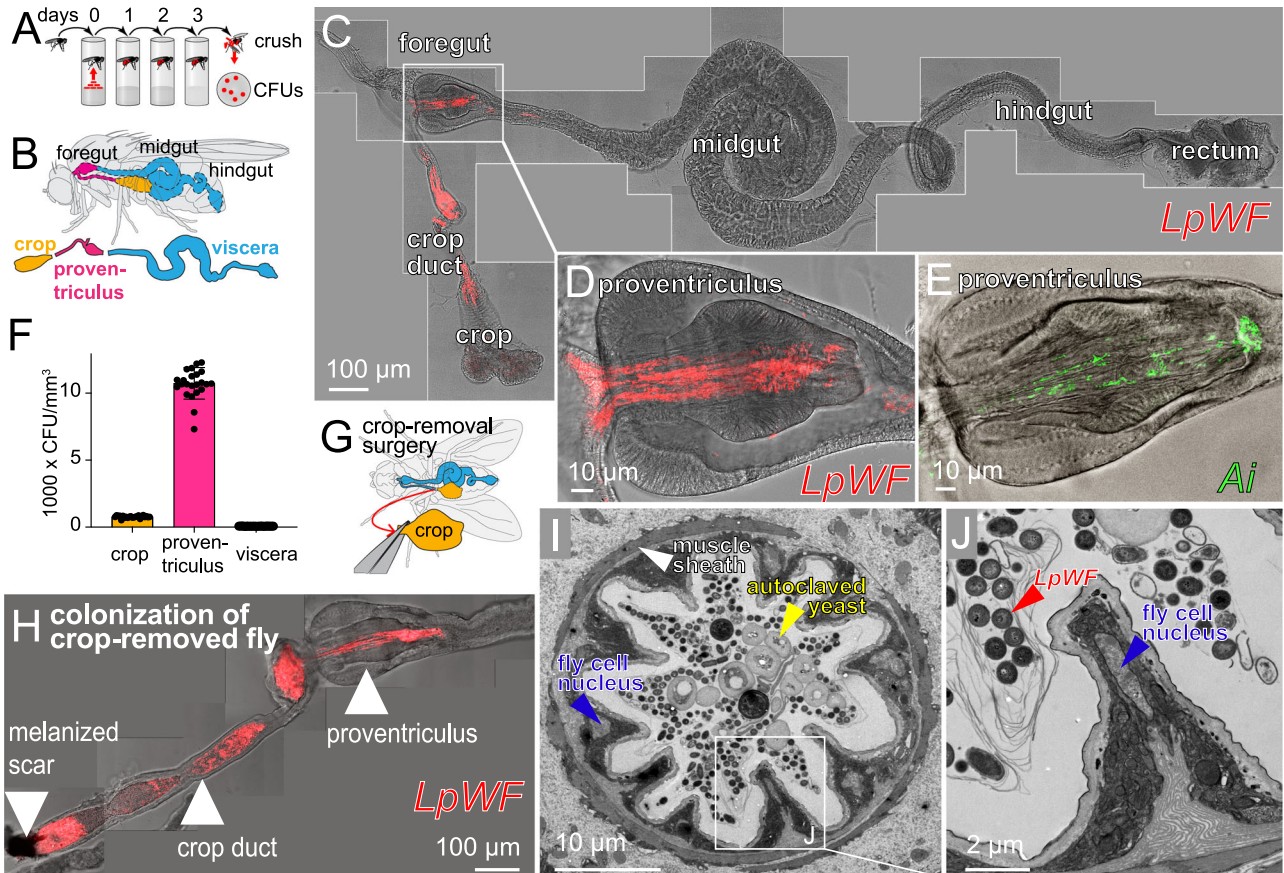

**Fig. 1 | *LpWF* stably colonizes the fly gut with spatial specificity. A** Colonization assay schematic with initial dosing on day 0 and serial transfers to sterile food daily for 3 days before analysis. **B** Gut diagram. **C** Microscopy of *LpWF*-mCherry colonization in a whole gut after clearing transient cells shows a specific colonization zone in the foregut. Shown is a maximum-intensity *z*-projection. **D** The proventriculus is a major site of *LpWF* colonization. **E** *Ai* colonization is also specific to the proventriculus lumen and crop duct (see also Fig. S2). **F** CFU densities from regions dissected in **B**. *n* = 23 individual guts/region from three biological replicates. Columns represent means. Error bars are S.D. **G** Microsurgery was carried out to remove the crop. **H** *LpWF* colonizes the foregut of flies with the crop removed (*n* = 15/15). **I** TEM cross-section of proventriculus inner lumen. Representative image of *n* = 3 biological replicates. **J** Detail of (**I**). Scale bars are defined in the figure panels. Source data are provided as a Source Data file.

*LpWF*-mCherry: ~$10^3$ *LpWF*-mCherry CFUs were required for 50% of flies to be colonized, while 100% of germ-free flies ended up colonized by doses as low as $10^2$ CFUs (Fig. 2D). These findings demonstrate that the proventricular niche for *LpWF*, when occupied, strongly resists colonization by later doses of the same strain.

The relationship between the probability of establishment and the final abundance of successful colonizing bacteria suggests that the availability of open habitats regulates the chance of invasion. We formalized assumptions of this hypothesis by building an integrated theory of initial colonization[24] and niche saturation[42] that predicts the likelihood of colonization, $P(N_0)$, of an invading species inoculated at a dose of $N_0$ as a function of the final abundance of the invading species, $A(N_0)$:

$$P(N_0) = (1 - p)^{A(N_0)/pk}, \tag{1}$$

where $p$ is the colonization probability of an individual bacterial cell and $k$ is the subpopulation size attained in a single successful colonization event (Fig. S5A, B). This model allowed us to estimate the scale at which the population is structured based on colonization probabilities and total bacterial abundances. For *LpWF*, Eq. 1 estimates a subpopulation size of $k$=600 cells (Fig. S5C), which is roughly the number of cells contained in an individual furrow.

To test whether the later dose of *LpWF*-mCherry was spatially excluded by resident *LpWF*, we constructed a GFP-expressing strain

of *LpWF* and fed it to flies pre-colonized with *LpWF*-mCherry. We imaged whole fixed guts 1 h post inoculation (hpi) to capture *LpWF*-GFP cells before they passed out of the fly (Fig. 2E). In the proventriculus, the invading *LpWF*-GFP were localized along the central axis of the inner lumen, separated from the lumen wall by a layer of resident *LpWF*-mCherry (Fig. 2E, F) that was up to 10 μm thick. The posterior proventriculus furrows were densely packed with *LpWF*-mCherry, while *LpWF*-GFP was largely absent from furrows, suggesting that these furrows are the sites of stable colonization. We confirmed that the fluorophores are not responsible for the differential colonization by feeding *LpWF*-mCherry to flies pre-colonized by unlabeled *LpWF* and quantifying the mCherry signal along the gut at 1 hpi and 24 hpi. At 24 hpi with a dose of ~$10^4$ CFUs, flies pre-colonized by *LpWF* showed almost undetectable mCherry by microscopy (Fig. S4E–H). These results provide further support that the niche for *LpWF* is in the proventricular furrows. Unlike during initial colonization, in which bacteria rapidly enter and colonize the furrows, prior colonizers prevented subsequent colonization, suggesting that there are a limited number of binding sites in the furrows for *LpWF* cells and that these sites are saturated by prior colonization. Consistent with this logic that niche priority is spatially determined, in the cases when *LpWF*-GFP did show colonization (*n* = 5), GFP-labeled cells were co-localized with each other along a furrow rather than being evenly mixed with mCherry (Fig. S4I).

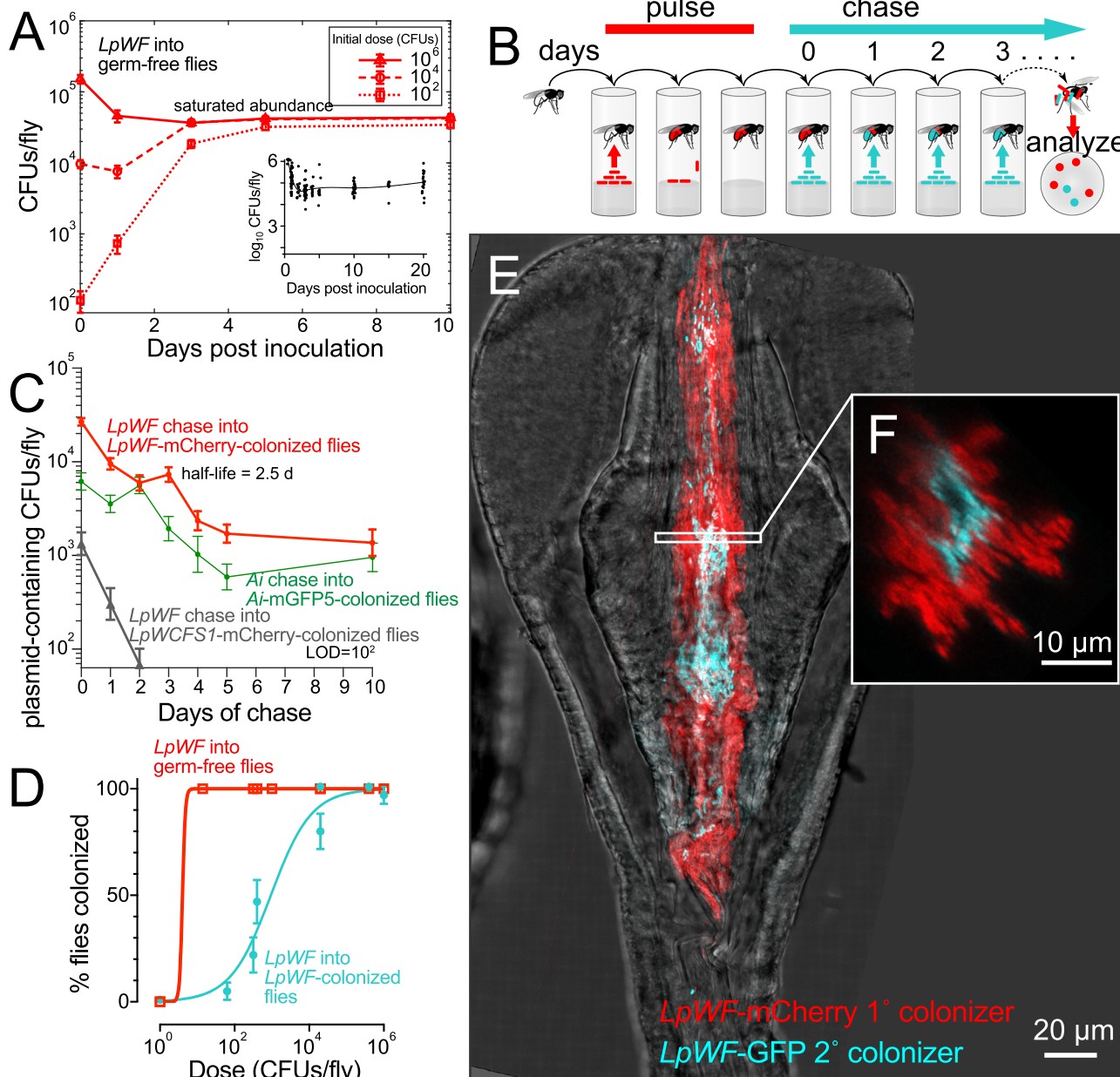

**Fig. 2 | Kinetic properties of bacterial association suggest the existence of a niche in the proventriculus. A** Saturation occurs over a colonization time course of germ-free flies by *LpWF*. Data points are mean of $\log_{10}$(CFUs) in $n \geq 48$ flies/data point. Error bars represent 1 s.e.m. Inset: 20-day time course after inoculation with $10^6$ CFUs (data from[24]). **B** Bacterial pulse-chase experimental design: flies were first pre-colonized with *LpWF*-mCherry, then fed an excess of unlabeled *LpWF* (blue) daily on fresh food. **C** Acterial cell turnover quantified by pulse-chase time course of *Lp*-mCherry-pre-colonized flies continuously fed unlabeled *LpWF* or *Ai*-GFP-pre-

colonized flies continuously fed unlabeled *Ai*. Data points are mean of $\log_{10}$(CFUs) in $n \geq 34$ flies/data point. Error bars represent 1 s.e.m. **D** Colonization efficiency quantified by dose response to colonization of individual flies. CFUs were measured at 3 dpi of the second colonizer. $n = 24$ flies/dose, error bars represent 1 standard error of the proportion. Limit of detection: 50 CFUs. **E** Spatial structure of colonization dynamics in the proventriculus for a fly pre-colonized with *LpWF*-mCherry (red) invaded by *LpWF*-GFP and imaged 1 h post inoculation (hpi). **F** Optical *x,z*-slice. Source data are provided as a Source Data file.

### *Ai* and *LpWF* occupy separate niches within the proventriculus

Interspecies interactions can have major impacts on ecosystem colonization through priority effects that include competitive exclusion and facilitation[43–47]. Because *Ai* and *LpWF* colonize the same general location of the gut (Figs. 1C–F, S3A–D) and each strain excludes itself (Figs. 2D, 3A), we expected that they would exclude each other. To test this hypothesis, we measured each species's abundance and growth rate during co-colonization. To our surprise, both were unaffected (Figs. 3B, C, S6), demonstrating that the species independently saturate the niche. We also performed a dose-response assay to determine whether interactions affect

establishment of new colonizers. By contrast to *Ai*'s self-exclusion, *LpWF* pre-colonization facilitated *Ai* colonization (Fig. 3A), while *LpWF* colonization was unaffected by the presence of *Ai* (Fig. S6A–C). *A. pasteurianus*, a phylogenetically distinct species of *Acetobacter*[48] that is common in *D. melanogaster*[49], was also facilitated by *LpWF* (Fig. S6D). Heat-killed *LpWF* did not facilitate *Ai* colonization, indicating live *LpWF* cells are necessary to facilitate *Ai* colonization (Fig. S7A, B).

Fluorescence microscopy of guts co-colonized by *LpWF*-mCherry and *Ai*-GFP showed that *Ai* and *LpWF* co-colonized the same foregut regions (Fig. 3D), with distinct sectors of each species

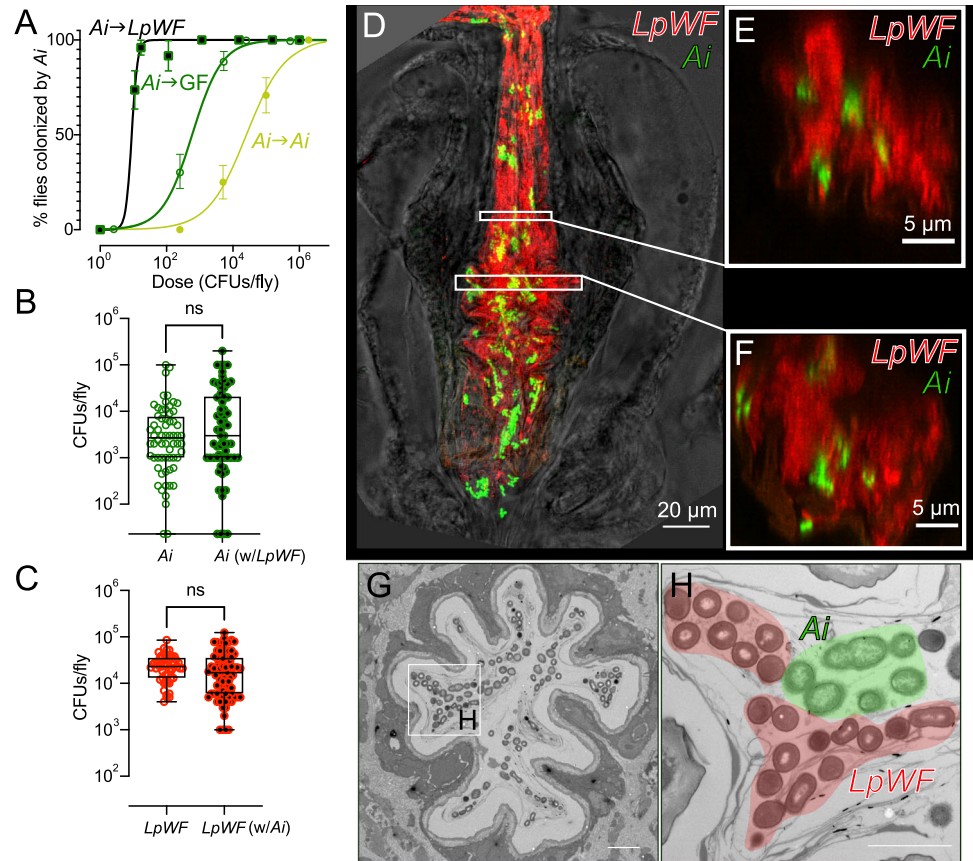

**Fig. 3 | *Ai* and *LpWF* occupy separate niches within the proventriculus. A** Strain interactions influence colonization efficiency, as seen by dose-response curves for *Ai* fed to germ-free flies (open green circles), *Ai*-pre-colonized flies (filled yellow circles), or *Lp*-pre-colonized flies (black-filled green squares). Z-test of differences in proportion versus *Ai* into germ-free flies: dose $10^{2.3}$ CFUs/fly, $p = 8.1 \times 10^{-4}$; dose $10^{3.7}$ CFUs/fly: $p = 4.8 \times 10^{-9}$; dose $10^5$ CFUs/fly: $p = 8.7 \times 10^{-6}$. $n \geq 12$ flies/data point. Error bars represent 1 standard error of the proportion. **B** *Ai* abundance at 5 dpi does not differ between flies mono-colonized with *Ai* versus pre-colonized with *LpWF* then fed *Ai*. $n \geq 65$ flies/treatment; two-tailed unpaired t-test, $p = 0.38$; ns indicates not significant. **C** *LpWF* abundance 5 dpi does not differ between flies mono-colonized with *LpWF* versus pre-colonized with *Ai* then fed *LpWF*. $n \geq 53$ flies/ treatment; two-tailed unpaired t-test; $p = 0.06$; ns indicates not significant. **B**, **C**: Center of box is median; box encloses 25th to 75th percentiles; whiskers indicate minimum and maximum. **D** Confocal microscopy of *Lp* and *Ai* co-colonization. *Ai* (green) and *LpWF* (red) occupied the same regions of the foregut 1 dpi. Scale bar: 100 μm. **E**, **F** *x,z*-section of *Ai* and *LpWF* sectors. **G** TEM cross-section of *Ai* and *LpWF* co-colonizing the anterior proventriculus. Scale bar: 5 μm. **H** Detail of (**G**) with *LpWF* and *Ai* cells pseudocolored. Scale bar: 2 μm. Source data are provided as a Source Data file.

(Fig. 3E–H, Supplementary Movie 1). Thus, *LpWF* and *Ai* do not physically exclude one another; instead, the tissue accommodates both strains.

### Colonization of the niche induces morphological alteration of the proventriculus

To examine the coexistence of overlapping *Ai* and *LpWF* populations in a physically confined space, we imaged fly anatomy using X-ray microcomputed tomography (XR μCT)[50,51], and segmented the volumetric image data to produce 3D reconstructions (Fig. 4A). We imaged germ-free flies and flies colonized with *LpWF*, *Ai*, or both *LpWF* and *Ai*. Numerous crypts were apparent along the length of the gut, including in uncolonized regions of the midgut and hindgut that are shielded by peritrophic matrix (Figs. 4A, S8)[52]. In the colonized region of the foregut, the longitudinal striations where we observed bacteria coincided with ridges and furrows of host tissue in the proventriculus inner lumen and crop duct (Fig. 4B–F). The furrows were straight in the anterior proventriculus, becoming larger and more irregular in the posterior (Fig. 4D, F). Transverse slices of the lumen wall revealed a narrow passage through the germ-free proventriculus (Fig. 4C), while the opening was much broader in the colonized proventriculus (Fig. 4E), corresponding to a significantly higher luminal volume than in germ-free flies (Fig. 4G).

Consistent with XR μCT imaging, TEM cross-sections of the proventriculus of germ-free flies showed a narrow luminal space, approximately 0.5 μm in diameter (Figs. 4H, S9). Similar morphology was observed in conventionally reared lab flies, which are associated with poor-colonizing strains of the same bacterial species, including *L. plantarum*[24] (Fig. 4I). In *LpWF*-colonized flies, the diameter of the furrows increased to ~1 μm by 1 hpi (Figs. 4J,K, S9) and ~2–3 μm by 3 dpi (Figs. 4L, M, S9E–J), suggesting a sustained host response to niche occupancy. Heat-killed *LpWF* did not produce this niche expansion, indicating live *LpWF* cells are necessary for the luminal expansion (Fig. S7C–F). By TEM, the expanded luminal space of the colonized proventriculus contained two zones: a clear zone adjacent to the lumen wall, and a bacteria-colonized zone closer to the center of the lumen (Fig. 4L, M, S9E–J). High pressure freezing fixation showed the same phenotypes (Fig. S9S), indicating that the zonation is not simply an artifact of fixation. Taken together, our imaging results show that the proventriculus undergoes morphological changes upon colonization, which coincide with the promotion of *Ai* colonization.

### Lectin staining reveals a glycan-rich matrix associated with the foregut niche

Mucus is heavily glycosylated, with various glycan subunits including *N*-aceytlglucosamine, *N*-acetylgalatosamine, *N*-acetylneuraminic acid,

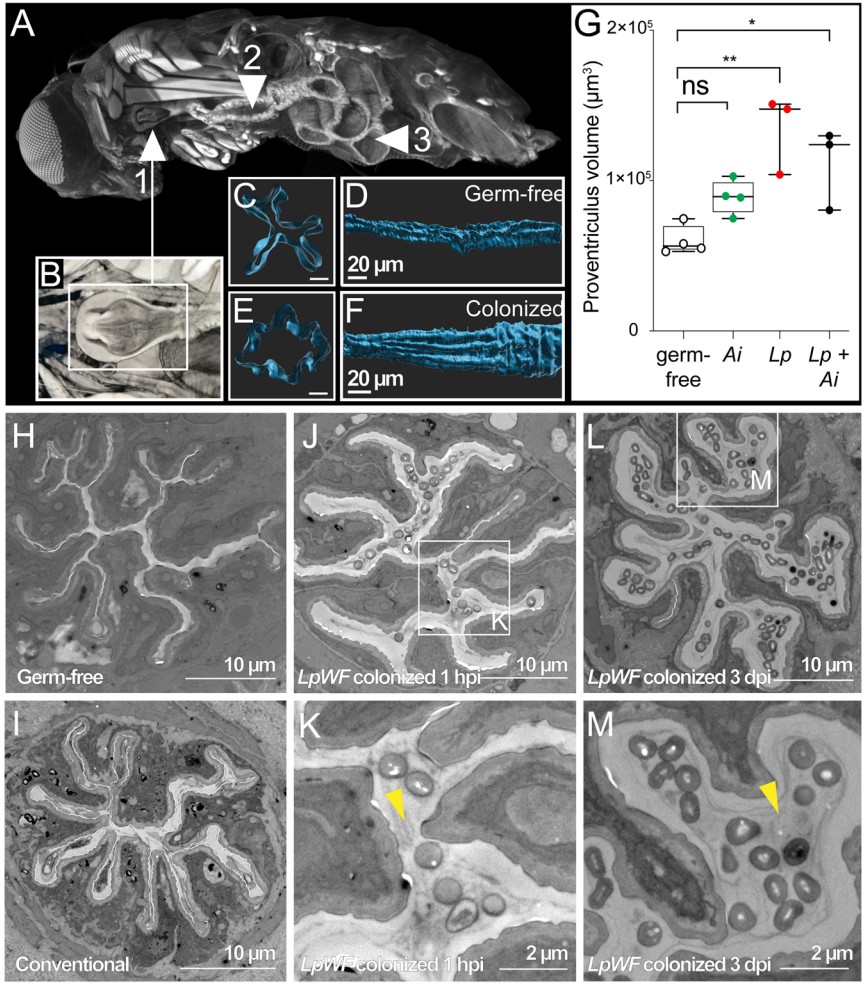

**Fig. 4 | Colonization of the niche induces morphological alteration of the proventriculus.** **A** Xray μCT model of a whole fly. Cutaway shows (1) exposed proventriculus (also inset of (**B**)), (2) anterior midgut, and (3) posterior midgut. **B** Detail of proventriculus. **C** Cross-section of a germ-free proventriculus inner lumen. Scale bar: 5 μm. **D** Germ-free proventriculus inner lumen volume rendering. **E** *LpWF*-colonized proventriculus inner lumen cross-section. Scale bar: 5 μm. **F** *LpWF* proventriculus inner lumen volume rendering. **G** Cardia volume calculated from surface models (*n* = 3 to 4 flies per condition; *p* = 0.0025, one-way ANOVA relative to germ-free; Tukey's correction for multiple comparisons; GF vs. *Lp* *p* = 0.020; GF vs. *Lp+Ai* *p* = 0.022.). **H–M** Transmission electron microscopy transverse cross-section of anterior proventriculus in (**H**) germ-free fly, (**I**) conventionally-reared fly (only lab fly bacteria; no *LpWF*), (**J**, **K**) 1 hpi with *LpWF*, (**L**, **M**) 3 dpi colonized with *LpWF* (see Fig. S7). *n* ≥ 3 biological replicates per treatment for TEM. Yellow arrowheads indicate lumen space. Source data are provided as a Source Data file.

mannose, glucose, fucose, and arabinose, depending on the type of mucus[53]. We hypothesized that the extracellular matrix (ECM) in the colonized zone of the proventriculus inner lumen, as seen by thin section TEM (Figs. 4, S9), is glycan-rich. To test this hypothesis, we sectioned the proventriculus and stained with a panel of lectins that have specificity for glycans found in mucus ("Methods"). Wheat germ agglutinin (WGA), *Dolichus biflorus* agglutinin (DBA), and *Lens culinarus* agglutinin (LCA) stained fly cells in the proventriculus sections, while other lectins showed no consistent binding to the proventriculus (Fig. 5A). Examining the inner lumen of the proventriculus where the secreted layer appears by TEM (Figs. 4H−M, S9), we observed staining by WGA (Fig. 5B). DBA and LCA staining was also present in the inner lumen but the locations did not correspond to the secreted layer seen by TEM (Figs. S10A, B). WGA binds *N*-acetylglucosamine and can also bind *N*-acetylneuraminic acid[54], the major sialic acid found in mammalian cells. *Sambucus nigra* agglutinin, which binds sialic acids, did not stain, consistent with published reports that sialic acid occurs only in fly embryos[55]. Succinylated WGA (sWGA) binds only *N*-acetylglucosamine[54] and exhibited a staining pattern consistent with WGA and TEM images of the secreted layer, indicating that the inner lumen ECM is *N*-acetylglucosamine rich (Fig. 5C). Chitin is a polymer of

*N*-acetylglucosamine, and a permeable, chitinous cuticle is also present in the proventriculus[38]. Calcofluor, which binds chitin, stained the cuticle, which can be seen clearly between the inner lumen epithelial cells and the secreted layer, indicating that the secreted layer is not chitin (Fig. 5B, C). TEM suggested germ-free flies also have a narrow secreted layer (Fig. 4H). We analyzed germ-free flies and found that this narrow layer stained with WGA, indicating that *N*-aceytlglucosamine is a primary glycan in the secreted layer of germ-free flies (Figs. 5D, S10C). To determine whether the layer could be the product of digested chitin from yeast cell walls in the food, we sectioned newly eclosed virgin females before their first feed and found that staining of their proventriculus was consistent with germ-free flies (Figs. 5E, S10D, E). Because the proventriculus is formed from an imaginal disk during pupariation, our evidence indicates the secreted *N*-acetylglucosamine-rich ECM is clearly produced by the fly.

## Discussion
Our results show that specific strains of *Drosophila* gut bacteria colonize crypt-like furrows in the proventriculus, that the colonization by these strains is saturable, suggesting a limited number of binding sites, and that the proventriculus responds to colonization through

**A**

| Lectin | Target | Inner lumen cells | Inner lumen | Secreted layer |
|---|---|---|---|---|
| WGA | N-acetylglucosamine, N-acetylneuraminic acid | + | + | + |
| S-WGA | N-acetylglucosamine | + | + | + |
| DBA | N-acetylgalactosamine | + | - | - |
| LCA | mannose, glucose | + | +/- | - |

**Fig. 5 | Lectin staining reveals a glycan-rich matrix associated with the foregut niche. A** Table of lectins that stained the proventriculus. Lectin staining of proventriculus transverse sections for **B** WGA in a colonized fly, **C** sWGA in a colonized fly, **D** WGA in a germ-free fly, and **E** WGA in a newly eclosed germ-free fly before first food ingestion. Scale bars: 20 μm. Arrowheads indicate lectin staining in the interior proventriculus lumen. $n \geq 3$ biological replicates per treatment. CF: calcofluor.

engorgement, which promotes colonization by bacteria that benefit the fly[32,33,56]. The finding that *Drosophila* has a specific niche for binding of commensals to sites in the crop duct and proventriculus is significant because it provides insight into how a microbiome can interact with the host in a manner that can be host-regulated and mutually beneficial. Furthermore, it predicts the existence of specific molecules in the extracellular matrix of the proventriculus that bind to the bacterial surface of colonization-competent strains but not to non-

colonizing strains. Though we have not investigated mucins here, flies have mucins[57,58], and three of these are expressed in the proventriculus[52]. Furthermore, the morphology of the extracellular matrix in the proventriculus is reminiscent of mammalian mucus, which has two layers: a dense, uncolonized layer adjacent to the epithelium, and a thinner, distal layer colonized by bacteria[59]. We speculate that the niche cells in the proventricular furrows either produce the adhesion substrate for the bacteria or produce an extracellular

matrix that sequesters the adhesion substrate from another source, such as the salivary gland. The apparent densely stacked membranes observed by TEM of the niche cells (Fig. 1J) suggests these may be secretory cells, which is consistent with the extensive secretory nature of the proventriculus[38]. The finding that binding of one bacterial strain can lead to structural changes that open niche sites for a second bacterial species provides a model for how complex assemblies of bacterial strains can arise and be maintained within a host digestive tract, with the host selecting primary colonizers that in turn select secondary colonizers. Indeed, *L. plantarum* is homofermentative, and we have previously shown that *LpWF* has a positive metabolic interaction with *Acetobacter* strains found in the fly gut, including *Ai*[60].

Despite the long history of studies on the *Drosophila* microbiome, the existence of a specific niche has been obscured by the presence of bacteria in the food and on traditional culturing media. A substantial fraction of gut bacteria under such conditions simply pass through and do not interact specifically with the gut[61], even though specific microbiome members bound to their associated niches might be present. We used bacterial pulse-chase protocols to push out unbound bacteria. This methodological advance enriched for only specifically interacting cells, allowing us to identify the symbiotic niche.

Possession of a microbiome is clearly highly beneficial for *Drosophila*, given that axenic flies show strongly reduced growth and fecundity[29–32,62,63]. However, it is less clear how the relationship between the host and specific strains of bacteria is stably perpetuated. A previous study showed that larvae excrete *N*-acetylglucosamine in their frass, providing a nutrient that aids external *L. plantarum* growth on fly food[37]. While the origin of the larval excreta *N*-aceytlglucosamine was not determined, our results indicate that the adult proventriculus niche is enriched in fly-produced *N*-aceytlglucosamine, suggesting that the niche provides both a spatial habitat and a nutritional source for *L. plantarum*.

Understanding the proventricular niche is likely to provide insight into microbiome function by (1) revealing the spatial locations where bacteria influence the host to introduce molecules into the gut, perhaps along with the peritrophic membrane; and by (2) revealing whether changes in niche structure induced by one species lay the groundwork for more complex associations between different members of the microbiome, such as *LpWF* and *Ai*, that are related to their functional pathways. Finally, these observations raise the question of whether additional niches exist at other locations in the *Drosophila* digestive system and within the gut of other animals, including humans.

# Methods

### Fly strains and rearing
No ethical approval was obtained because insect models do not require ethical approval under local laws and regulations. All flies in this study were mated females (except in Fig. S2), which show low heterogeneity in gut morphology[40]. Previous work showed that the colonization phenotypes we measured are general across multiple genetic backgrounds, including CantonS, *w1118*, and OregonR[24]. Flies were reared in Wide Drosophila Vials (Cat #: 32-114, Genesee), with Droso-Plugs® (Cat #: 59-201, Genesee). Food composition was 10% glucose (filter-sterilized), 5% autoclaved live yeast, 0.42% propionic acid (filter-sterilized), 1.2% autoclaved agar, and 0.5% autoclaved cornmeal. Each vial contained 4 mL of food. Germ free fly stocks were passaged to fresh vials every 3–4 d. Five day-old mated female adults were sorted the day prior to beginning an experiment.

Liquid food was composed of 10% glucose, 5% yeast extract, and 0.42% propionic acid. The only nutritional difference between liquid and solid food was yeast extract instead of autoclaved live yeast because the yeast cell walls clog the capillaries used for liquid feeding. The bottom of capillary feeder vials contained 1.2% agar as a hydration and humidity source. Both CAFÉ- and solid food-fed flies were

transferred daily to fresh vials to minimize bacterial re-ingestion. Samples of flies were surface-sterilized and crushed, and CFUs were enumerated.

### Bacterial strains
Bacterial strains were reported in ref. [24], including *Lactobacillus plantarum WF*, *L. plantarum LF*, and *L. plantarum WCFS1*, which was called *L. plantarum HS* in ref. [24]. *Acetobacter indoneiensis SB003* was assayed for colonization in Fig. S1 of ref. [24]. Fluorescent protein-expressing plasmid strains were developed and reported in refs. [24,60]. pCD256-p11-mCherry, used for *L. plantarum*, was the generous gift of Reingard Grabherr (BOKU, Austria)[64]. pCM62, used for *Acetobacter indonesiensis*, was the generous gift of Elizabeth Skovran (SJSU, USA).

### Colonization assay
The colonization assay followed the protocol used in Fig. S1A of ref. [24]. Briefly, a measured dose of bacteria was pipetted evenly on the surface of a germ-free fly food vial and allowed to absorb for 15 min. Twenty-five germ-free, 5- to 7-d post-eclosion, mated female flies were introduced to the vial and allowed to feed for a defined period of time. Flies were then removed from the inoculation vial and placed in fresh, germ-free vials. Bacteria were collected from the inoculation vial by vigorous rinsing with PBS, and the abundance was quantified by CFUs. At specified time points, CFUs in individual flies were enumerated by washing the flies 6 times in 70% ethanol, followed by rinsing in ddH$_2$O, and then crushing and plating for CFU enumeration.

### Preparation of bacteria
Cultures of bacteria were grown overnight in 3 mL liquid media at 30 °C. *Lp* strains were grown in MRS liquid media (Hardy Diagnostics, #445054), and 10 μg/mL chloramphenicol was added for mCherry-expressing strains. *Ai* was grown in MYPL media, and 25 μg/mL tetracycline was added for GFP-expressing strains. Bacteria were pelleted by spinning for 3 min at $400 \times g$, resuspended in PBS, then diluted to the desired concentration. Dose size was quantified using OD$_{600}$ or by plating and counting CFUs. OD of 1.0 corresponds to $2 \times 10^8$ CFUs/mL for *LpWF* and $3 \times 10^8$ CFUs/mL for *Ai*.

### Inoculation of flies
Flies were inoculated by pipetting 50 μL of an appropriate concentration of the inoculum onto the food and then left to dry in the biosafety cabinet for 15 min. Flies were starved for 4 h before flipping them into the inoculation vials, where they were allowed to feed for 1 h, then flipped to fresh vials. The dose per fly was calculated as the amount of inoculum consumed divided by the number of flies in the vial. To verify that flies ate the bacteria placed on the food and measure the amount of ingested inoculum, uneaten bacteria were recovered from the vial after feeding and subtracted from the original dose. For experiments to standardize the dose of bacteria, we used an inverted 50-mL conical vial with solidified agar food in the cap, which allows for separation of food CFUs from CFUs on the walls of the vial. For other experiments, we used an autoclaved, polypropylene wide fly vial (Genesee).

### Larvae colonization
Larvae were colonized by first inoculating food vials with 100 μl of bacterial culture at an OD of 1. At least 25 mated female flies were added to the vials and allowed to lay eggs for 1 day, then removed. 3 days later, 3rd instar larvae were collected and washed in PBS, then transferred to sterile agar-water vials for 4 h to clear transient bacteria. Larval guts were dissected whole, then mounted in mounting media (80% glycerol, 20% Tris 0.1 M pH 9.0), then imaged using a confocal microscope to capture a *z*-stack of the entire proventriculus.

## Quantification of CFUs in flies

Abundance in the gut was measured by homogenizing whole flies then plating to count CFUs. Flies were first anesthetized using $CO_2$ and surface-sterilized by washing twice in 70% ethanol, then twice in PBS. Next, they were placed individually into wells of a 96-well plate along with 100 μL of PBS and ~50 μL of 0.5-μm glass beads (Biospec) and heat-sealed (Thermal Bond Heat Seal Foil, 4titude). The plate was shaken violently for 4 min at maximum speed on a bead beater (Biospec Mini-beadbeater-96, #1001) to homogenize the flies. We previously showed that the 0.5 μm bead size does not diminish bacterial counts and effectively disrupts fly tissue[24]. A dilution series of the entire plate was prepared using a liquid-handling robot (Benchsmart). Agar growth medium was prepared in rectangular tray plates, which were warmed and dried ~30 min prior to plating. Plates were inoculated with 2 μL of fly homogenate per well, which leads to a circular patch for CFU enumeration. The plates were incubated at 30 °C overnight. To count colonies, plates were photographed under fluorescent light and counted semi-automatically using ImageJ 2.1.0[65].

## Measurement of CFUs in fly vials

The number of bacteria in a fly vial was measured by recovering cells from the vial and plating on nutrient agar growth media (MRS or MYPL) to count CFUs. To collect bacteria, 2 mL of sterile PBS were pipetted into the vial. The vial was then replugged and vortexed for 10 s. A dilution series was made starting with 100 μL of the PBS wash and then plated to count CFUs. This method was used to quantify viable bacteria egested (defecated) by flies, or bacterial growth in the vial or the remainder of uneaten inoculum. Egestion and inoculation were measured over a period of 1–2 h, minimizing the opportunity for new bacterial growth.

## CAFÉ assay

Twelve flies were placed in a sterile polypropylene wide mouth fly vial containing 2 mL of 1.2% agar in ddH$_2$O. Four glass capillary tubes were inserted through the flug and filled with 12 μL of filter-sterilized liquid fly food (10% glucose, 5% yeast extract, 0.42% propionic acid). Ten microliters of overlay oil were added on top to push the liquid food to the bottom of the capillary. Flies were left in the vial for 24 h before being transferred to a fresh setup. Vials were checked every 12 h to ensure flies had access to food, and a fresh flug with new capillaries was inserted if capillaries had air in them, which prevents food access. Five fly vials were put together into a 1-L beaker with a wet paper towel at the bottom and aluminum foil over the top, and the beaker was placed in the back of a fly incubator set to 25 °C, 12 h-12 h light-dark cycling, and 60% relative humidity.

## Heat killed *LpWF* treatment and dose response

*LpWF* bacteria were prepared by growing an overnight culture in MRS + 10 μg/mL chloramphenicol at 30 °C. The culture was pelleted by centrifugation at 400 × $g$ and then resuspended in PBS at an OD of 2. To kill the bacteria, 1 mL of the resulting suspension in a 1.5 mL microcentrifuge tube was heated to 65 °C for 30 min in a USA Scientific Mini Dry Bath. A sample of the suspension was spread on an MRS plate and incubated for 2 days to confirm that all cells were successfully killed.

To simulate colonization by live *LpWF*, flies were fed with excess heat-killed bacteria daily for 3 d. 5- to 7-day old mated female flies were kept at 25 flies/vial, as in the other colonization experiments. Each day, 100 μL of the heat-killed suspension ($4 \times 10^7$ CFUs/vial) was pipetted onto fresh, sterile food and allowed to dry before transferring flies into the vial. In order to clear excess killed bacteria from the niche before performing experiments to evaluate the impact on the host, flies were transferred to fresh sterile food overnight and then further cleared by placing on agar-water media for 4 h before imaging or dosing with *Ai*.

From this point on, the methods were the same as if flies were colonized by our standard method.

## Pulse-chase protocol for bacterial colonization

To estimate the turnover time of established bacterial populations, 5- to 7-day old mated female flies were kept with 25 flies/vial. Flies were first inoculated with a pulse of fluorescently labeled, antibiotic-resistant bacteria by pipetting 50 μL of culture resuspended in PBS ($OD_{600} = 1$) onto the food and allowing it to dry prior to flipping flies into the inoculation vial. The pulse dose was allowed to establish colonization in the gut for 3 d prior to chase. Flies were fed a chase dose in the same way each day for 10 d ($OD_{600} = 1$). The abundance of labeled resident was measured daily by homogenizing and plating a sample of flies on selective media to count CFUs. The invading chase dose was assayed by plating on non-selective media. To control for any other factors that might affect resident abundance, a control group was also passaged daily to fresh food with no chase dose and assayed daily to count CFUs.

## Pulse-chase analysis

Measurements from individual flies from the different experiments were pooled by time point. Data were fit to an exponential decay using Prism, and the half-life with its confidence interval was reported.

## Measurement of growth rates in vivo

Plasmid loss in the absence of selection was used as a proxy for bacterial growth rate[24]. Briefly, a standard curve was constructed by passaging plasmid-containing cells in fresh medium twice daily in a ~1:100 dilution to an OD of 0.01 for 6 d. The number of bacterial generations was estimated by counting the number of CFUs in the culture prior to dilution. The ratio of plasmid-containing CFUs to plasmid-free CFUs was counted as the number of fluorescent to non-fluorescent colonies. The doubling time is roughly 2 h for each strain. A linear regression was used to fit the standard curve data. Flies were then fed 100% plasmid-containing cells. The ratio of plasmid-containing to plasmid-free CFUs was counted at various time points, and the standard curve was used to convert the ratio to the number of doublings. In the case of dual-plasmid containing strains (Fig. S7C), growth was measured as a ratio of colonies positive for GFP-Erm plasmids (which are lost rapidly) to those positive for mCherry-Cam (which is retained much longer). A non-linear (exponential decay) regression was used. Two caveats are that (1) population bottlenecks cause wider variance in the plasmid ratio, and (2) in vivo plasmid loss rates may be different from in vitro rates. We previously showed that the first caveat can be used to infer bottlenecks. We also note that with respect to the second caveat, our use of this method to compare growth rates in a controlled experiment does not necessitate an absolute growth measurement with a standard curve. Furthermore, the growth rates in vivo were similar to in vitro, meaning that any differences in plasmid loss rates due to differences in the growth phase of the cells are likely small.

## Cropectomy

Cropectomy was performed on live flies using only new, undamaged fine forceps (#5, Dumont). Forceps, flypad, and microscope area were cleaned with 70% ethanol. Five- to 10-day old female flies were first anesthetized using $CO_2$ then placed on a depression slide for surgery. The fly was positioned on its back, and while holding the torso with one set of forceps a small puncture was made in the abdomen just below the thorax as shown in Fig. 1O. Pressure on the forceps was released slightly to allow the tips to open up, then grab onto the crop and pull it out through the puncture. If the crop duct was still attached, it was severed along the edge of a forceps. Flies were placed in a sterile food vial and given at least 3 d to recover. Survival rate varied by operator from 1 of 10 flies to 2 of 3 flies.

## Preparation of samples for microscopy

Whole guts were removed from the fly by dissection with fine forceps (Dumont). Tissue was fixed in 4% PFA in PBS for 3 h at 24 °C or at 4 °C overnight. Guts were permeabilized using 0.1% Triton-X in PBS for 30 min at room temperature, washed twice in PBS, stained with 10 µg/mL DAPI for 30 min, washed twice in PBS, placed in mounting medium for up to 1 h, then transferred to the slide using a wide-bore 200 µL pipette. Each gut was then positioned on a positively charged glass microscope slide, and approximately 60 µL of mounting medium were added (mounting medium: 80% glycerol, 20% 0.1 M Tris 9.0, 0.4 g/L N-propyl gallate). Five to ten 0.1 mm glass beads (Biospec) were added to the mounting medium to form a spacer that prevents crushing of the sample. The slide was then covered with a No. 1.5 cover glass and sealed with nail polish.

## Confocal microscopy

Microscopy was conducted with a Leica DMi8 confocal microscope using either a 40× (1.30 NA) HC Plan Apo or a 60× (1.40 NA) HC Plan Apo oil immersion objective. Laser lines were generated using a white-light laser with AOTF crystal, and excitation wavelengths for fluorophores were: mGFP5, 488 nm; mCherry, 591 nm; Cy5, 650 nm. Whole gut images were generated by tiling multiple captures then merging using the Mosaic Merge function in the Leica Application Suite LAS X to stitch into a single stack. Z-stacks for whole guts were 70–80 µm in thickness with slices every 0.5 µm or less. To render two-dimensional images for publication, fluorescence channels were processed as maximum intensity z-projections and the brightfield channel is represented by a single z-slice from the middle of the stack.

## Spatial quantification of colonization

The spatial distribution of gut colonization was quantified based on microscopy images using FIJI to mask and segment the gut regions from the images and MATLAB to quantify the extent of colonization. First, in FIJI, summed intensity z-projections of 80 µm optical sections were generated, then resized to a scale of 1 µm/px. Background subtraction with a rolling ball radius of 50 px was applied. Equivalent measurements were made on germ-free guts to quantify the autofluorescence of the gut, which varies by region.

Next, a segmented line using a spline fit and a variable width of 40–200 µm depending on the gut width of the particular region (e.g., 200 µm for the wide part of the crop and 40 µm for the proventriculus) was drawn along the length of the gut, starting with the most distal point on the crop as the origin. The "Plot Profile" function was used to measure the intensity along each of 11 segments: crop (2 segments), crop duct (2 segments), crop duct-proventriculus junction (1 segment), proventriculus (1 segment), midgut (3 segments), and hindgut (2 segments). These intensity profiles were then exported to MATLAB.

In MATLAB, the segment length for each gut region was calibrated to the average length of each segment using a bilinear fit. This step resulted in all the guts being aligned and having the same overall size so that we could compare the intensity for each region of the gut across the replicate guts. The value of each intensity value along the spline was then background subtracted and normalized to the intensity of the visually confirmed bacteria in that region. This step adjusts for the variation in autofluorescence along the gut (highest in crop and hindgut), and it adjusts for the differences in fluorescence intensity of the bacteria due to different tissue depths. After this normalization step, a 100 µm moving average filter was applied within the boundaries of each gut region to smooth the small scale spatial variation. To convert each spatial location along the gut from the intensity of colonization to colonized/uncolonized, we thresholded the intensities of each gut based on visual inspection. The proportion of all the guts with colonization at each location along the spline was then plotted in the figure panels. Additionally, the number of guts with >5% colonization within each of the delineated regions is reported as a

percentage for each of the regions as follows: crop, crop duct, proventriculus, midgut, hindgut.

## Measurement of bead egestion

To measure shedding of polystyrene beads (Spherotech FP-0552-2, sky blue), flow cytometry was used to quantify the number of egested beads. Flies were kept in inverted 50 mL conical tubes with 1 mL of solid food in the cap. To collect shed material, the tubes were rinsed with 10 mL of PBS, vortexed for 10 s, and then a clean cap was placed on top. To concentrate the solution, the samples were spun in a centrifuge for 7 min at 400 × g. The pellet was then resuspended in 200 µL of PBS. The concentrated sample was counted on an Attune flow cytometer (Thermo Fisher).

## Electron microscopy

Whole guts were dissected in Cacodylate pH 7.4 (Cac) buffer, then fixed for 2 d in 3% GA + 1%FA in 0.1 M Cac at 4 °C. Samples were embedded in agarose and stored at 4 °C until further processing. Samples were then washed in Cac buffer, stained with 1% $OsO_4$ + 1.25% KfeCN for 1 h, washed in water, treated with 0.05 M maleate pH 6.5 (Mal), stained with 0.5% uranyl acetate in Mal for 1.25 h, then washed with increasing concentrations of ethanol. For embedding in resin, samples were treated with resin+propylene oxide (1:1) evaporated overnight as a transition solvent prior to embedding, then embedded in epoxy resin (Epon+Quetol (2:1)+Spurr (3:1)+2% BDMA overnight at 55 °C and cured at 70 °C for 4 d.

## X-ray micro-computed tomography (XR µCT)

Samples were prepared for XR µCT following the protocol in ref. [51], which the authors generously shared prior to publication. Briefly, flies were washed in 1% Triton-X in PBS to reduce cuticular wax. A shallow hole was poked in the abdomen and thorax with a fine tungsten pin to increase permeation of fixative and stain. Fixation was with Bouin's solution. Staining was with phoshotungstic acid for 3 weeks. Flies were mounted for imaging in a 10 µL micropipette tip containing deionized water and sealed with parafilm. Imaging was performed at the Lawrence Berkeley National Laboratory's synchrotron Advanced Light Source on beamline 8.3.2 with assistance of Dula Parkinson. 1313 images were acquired per specimen at 20× magnification through 180 degrees of rotation. Back-projections were performed using Tomopy with the following specifications:

*doFWringremoval 0 doPhaseRetrieval 1 alphaReg 0.5 doPolarRing 1 Rmaxwidth 30 Rtmax 300*

Further specifications are available here: http://microct.lbl.gov/. The images in Fig. 4A, B were produced in Octopus 8.8.2.7 and VG Studio 2.2. Volumetric reconstructions of the gut lumen in Fig. 4D–G were performed in Imaris using manual segmentation.

## Cryosectioning

To embed flies in OCT (McKessen, 981385), the legs and wings were first removed. Next, flies were equilibrated to the OCT medium by submerging and agitating to remove bubbles. Flies were then transferred to fresh OCT in the Cryomold (Ted Pella Inc., 4565, Lot 78652). Up to 5 flies were oriented in parallel in a single mold. The block was rapidly frozen by placing it on a bed of powdered dry ice and then stored at −80 °C until sectioning. 10 µm sections were prepared at −24 °C using a Leica CM3050 cryostat and transferred onto positively charged slides (VWR Superfrost Plus, 48311-703). Sections were air dried using a Mini Dry Bath (USA Scientific, BSH200) then stored at −20 °C until staining.

## Lectin staining

Sections were fixed and stained within 2 days of sectioning. First, excess OCT was removed from the slides by briefly washing in HBSS (Cold Harbor Springs Protocol). Next, sections were fixed in 4% PFA for

30 min, then washed once in HBSS. Before staining, the sections were blocked using 1% BSA in HBSS. Sections were stained for 30 min with the specific lectin at its optimal concentration: (12.5 μg/mL Wheat germ agglutinin (WGA – Biotium, 29026-1, Lot 18W1205), 50 μg/mL *Dolichos biflorus* agglutinin (DBA – Glycomatrix, 21511013-1, Lot L20092804ZH), 50 μg/mL *Lens culinarus* agglutinin (LCA – Glycomatrix, 21511020-1, Lot L20092902ZH), or 50 μg/mL WGA-Succinylated (S-WGA – Vectorlabs, FL-1021S-5, Lot 2008145). Lectin staining was performed simultaneously with the counterstain Calcofluor (Sigma-Aldrich, 910090, Lot MKCL1227) in HBSS. Sections were then washed once in HBSS and once more in 10% HBSS. Sections were mounted in mounting media (80% glycerol, 20% Tris 0.1M pH 9.0) and covered with a #1 cover slip.

The optimal concentration for staining of each lectin was determined by preparing concentrations of 50 μg/mL, 12.5 μg/mL and 3 μg/mL in the presence of the specific haptenic sugar at a concentration of 500 mM. For WGA and S-WGA, the haptenic sugar was N-acetylglucosamine (Vectorlabs, S-9002, Lot ZJ022). For DBA, the haptenic sugar was N-acetylgalactosamine (Vectorlabs, S-9001, Lot ZJ0301). For LCA, the haptenic sugar was D-mannose (Sigma, M602025G, Lot SLBG0980V). The specificity of each lectin was assessed by the inhibition of staining by the haptenic sugar. The lowest lectin concentration that stained the tissue and could be inhibited by its corresponding haptenic sugar was chosen as the optimal concentration and used for subsequent procedures.

### Statistics

Statistical tests were performed in Prism. In general, data were checked for normality using a Shapiro–Wilk test. If normality was established, a Welch's t-test was performed. Statistical tests of CFU abundances were performed on $\log_{10}$-transformed data. When CFUs were 0, the log was set to 0 (corresponding to a pseudocount of 1). When multiple comparisons were made, an ordinary one-way ANOVA was performed. If significant, multiple pairwise comparisons were performed with Tukey's multiple comparisons test. When data were not normally distributed, comparisons were made using Wilcoxon rank-order tests. Error bars on proportions are either standard error of the proportion (s.e.p.), or binomial 95% confidence intervals using the Clopper–Pearson method or Jeffries method, as specified in the text. The statistical significance of differences in proportions was assessed using a Z-test.

### Reporting summary

Further information on research design is available in the Nature Portfolio Reporting Summary linked to this article.

## Data availability

All data generated or analyzed during this study are included in this published article (and its supplementary information files). Source data are provided with this paper.

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

## Acknowledgements

Plasmid pCD256-mCherry[64] was generously provided by Reingard Grabherr of BOKU, Austria. Plasmid pCM62[66] was generously provided by Elizabeth Skovran of San Jose State University, USA. KCH is a Chan-Zuckerberg Biohub Investigator. Funding was provided by the Banting and the Pacific Institute for the Mathematical Sciences Postdoctoral Fellowships to EWJ, a Howard Hughes Medical Institute International Student Research fellowship, Stanford Bio-X Bowes fellowship, and the Siebel Scholars program to AA-D, the Allen Discovery Center at Stanford on Systems Modeling of Infection to KCH, the David and Lucile Packard Foundation and the Institute for Collaborative Biotechnologies through Grant W911NF-09-0001 from the US Army Research Office to JMC, a Natural Sciences and Engineering Research Council of Canada Discovery Grant and the Canada Research Chairs program to DAS, the Howard Hughes Medical Institute to ACS and CW, National Institutes of Health grants DP5OD017851 and R01DK128454 to WBL, National Science Foundation grant IOS 2032985 to WBL and KCH, National Science Foundation grant IOS 2144342 to WBL, the Carnegie Institution for Science Endowment to

WBL and ACS, a Carnegie Institution of Canada grant to WBL and DAS, and National Institutes of Health training grant T32GM007231 to KA. Work by ELB and MV was supported by the Laboratory Directed Research and Development Program of Lawrence Berkeley National Laboratory under U.S. Department of Energy contract No. DE-AC02-05CH11231.

## Author contributions

R.D., E.W.J., H.Z., B.O., C.W., E.B., J.M.C., D.A.S., A.S., and W.B.L. designed the research. R.D., E.W.J., H.Z., B.O., C.W., K.A., A.A.-D., M.V., and W.B.L. performed the research. R.D., E.W.J., D.J.M., and W.B.L. analyzed the data. R.D. and W.B.L. wrote the manuscript. R.D., E.W.J., K.C.H., D.A.S., J.M.C., A.C.S., and W.B.L. revised the manuscript. All authors reviewed the manuscript before submission.

## Competing interests

The authors declare no competing interests.
