## [Peer Review File · Nature Communications]

A symbiotic physical niche in *Drosophila melanogaster* regulates stable association of a multi-species gut microbiotaREVIEWER COMMENTS

Reviewer #1 (Remarks to the Author):

In this manuscript Dodge et al. study how intestinal bacteria colonize the gut of their animal host. Using *Drosophila* as a model host, the authors elegantly and with great depth demonstrate that intestinal bacterial colonize the adult foregut via priority effects and niche construction. This is an important and long-awaited characterization of a phenomena that has been heavily debated in the field albeit previously reported in papers from this lab (Obadia et al 2017) and Pais et al. 2018. The new work is compelling, very conclusive and of the highest standard of execution in the field.

A big positive aspect of the work is the unprecedented depth in the description of the phenomena which will be a great asset for the field however the concept of priority effect and niche construction was previously established in their previous work (Obadia et al. 2017) and the one of Pais et al. 2018 therefore, I am questioning the impact of the study for the large audience of *Nature Communications*. Yet, I leave it to the editor to judge this point as I very much like the work which is very robust and again I insist with an unprecedented depth of analysis.

Major points:

- I believe the author should report whether such residency, priority effect and niche construction also occur in virgin females, in males and in larvae.
- Beyond this, providing further mechanistical insight on the strain specificity of the colonization phenotype or hints on the mechanism underlying niche construction remain the outstanding questions that this referee would like see addressed (at least partly) in a high impact publication.

Specific points:

- Previous publications are not referenced properly in the supplementary materials file and currently cannot be matched with the main document references.
- FigS1K would benefit from having similar quantification as seen in FigS1J.
- line 130: I believe FigS2G should be referenced instead of FigS2F

Reviewer #2 (Remarks to the Author):

The manuscript by Dodge et al. presents a good body of work characterizing a physical niche in the *Drosophila* gut occupied and shaped by efficient bacterial colonizers. The data are solid in showing the different types of ecological interactions between initial and secondary colonizers, analogous to the notions colonization resistance and facilitation established in the mammalian microbiome studies. The paper is well written and the knowledge is valuable to the microbiome field.

Some comments:

- 1) L31-32: The significance of the study on host genetics and discovery of new niches in animals is over-extended and lacks support in the main text.

2) The proventriculus morphological change seems to be specifically dependent on LpWF and happen as quick as 1hpi. The authors suggest this is due to host response to some unknown bacterial molecules, not simply physical pressure. The authors also point out a mucus-like zone between the intestine wall and the lumen upon LpWF colonization. The reviewer finds it curious and valuable to provide at least some insights into the host response that leads to the mucus-like zone. To see if the host response is dependent on LpWF cell wall or active metabolism, one simple experiment would be to examine luminal space expansion upon 1hpi of heat-killed LpWF. How chemically similar are the gut mucosa between *Drosophila* and animals? An RNAseq experiment shortly after LpWF introduction would gain further mechanistic insights into the host response to niche occupancy.

3) Grammar: L68 an physical niche  a physical niche; L180 co-localized with each other a furrow  co-localized with each other in a furrow

Reviewer #3 (Remarks to the Author):

Dodge et al report that a physical niche regulates stable association of a multispecies gut microbiota. The researchers use germ-free, and gnotobiotic *Drosophila*, which are techniques that have been widely used in many labs over the past decade, due to the comparative straightforward methodology of generating germ-free flies. This model has been widely used for the mono-association, or gnotobiotic colonization of a known cocktail of bacteria. One strength of the manuscript is the high-quality imaging of the *Drosophila* whole alimentary canal. These techniques have continued to develop since some of the initial detailed characterization of physiological compartmentalization within the *Drosophila* midgut (PMID: 23991285). Using bacterial strains that are tractable within the *Drosophila* intestine is also a strength. However, despite these powerful imaging methodology, the biological conclusions drawn from the experiments only represent modestly advancements to the field. Due to the reductionist nature of the model, with only a couple of bacteria tested, certainly the title of the manuscript is rather overreaching in its conclusion, especially extrapolating to infer "multispecies gut microbiota". These interactions might be of *Lactobacillus* and *Acetobacter* might be highly specific to this model. Further examples of bacteria that physically remodels and intestinal niche to favor secondary colonization by an unrelated commensal would be required to make such generalizable conclusions.

REVIEWER COMMENTS (*italics*) and REBUTTALS (*indented*)

Reviewer #1 (Remarks to the Author):

In this manuscript Dodge et al. study how intestinal bacteria colonize the gut of their animal host. Using drosophila as a model host, the authors elegantly and with great depth demonstrate that intestinal bacterial colonize the adult foregut via priority effects and niche construction. This is an important and long-awaited characterization of a phenomena that has been heavily debated in the field albeit previously reported in papers from this lab (Obadia et al 2017) and Pais et al. 2018. The new work is compelling, very conclusive and of the highest standard of execution in the field. A big positive aspect of the work is the unprecedented depth in the description of the phenomena which will be a great asset for the field however the concept of priority effect and niche construction was previously established in their previous work (Obadia et al. 2017) and the one of Pais et al. 2018 therefore, I am questioning the impact of the study for the large audience of Nature Communication. Yet, I leave it to the editor to judge this point as I very much like the work which is very robust and again I insist with an unprecedented depth of analysis.

We thank the reviewer for appreciating the importance and high quality of this work. We believe that this work goes well beyond the original description of stable and specific colonization in Obadia et al 2017 and any speculation that it *might* represent a niche previously. Here we have experimentally dissected a host-regulated physical niche in what the reviewer called a "*compelling, very conclusive*" proof of the phenomenon at the "*highest standard of execution in the field*" with "*an unprecedented depth of analysis.*" Speculating that something might be true is quite different from documenting that it is true. Likewise, the work's impact is not reduced because it only documents specific binding with the properties of a niche for two species of bacteria: previously, there were zero species with highly documented, convincing, niche-like interaction.

To address the experimental concerns the reviewer brings up, we have some new information on these questions presented below. But we hope it is clear that the significance of the niche interactions we studied is not affected by whether or not the same interaction also takes place in virgin females, in males, or in larvae. It would still be a powerful model of specific interaction of a microbiome member with a host-regulated niche if only mated females possessed the niche. The high impact of our findings does not depend on these other questions.

Major points:

-I believe the author should report whether such residency, priority effect and niche construction also occur in virgin females, in males and in larvae.

We appreciate, based on the work of others in the field, that sex differences and mating can greatly influence the physiology of the gut. As the reviewer suggested, we tested whether the colonization phenotypes we observed—namely

initial colonization and colonization facilitation—occur in males, virgin females, and larvae. We found that males and virgin females do have the niche while larvae lack it. Since the larval proventriculus is constructed by a separate cell population from the adult one, the lack may reflect different cell types and extracellular secretions, which would be investigated in future work. The data is a new Figure S2 (see lines 87-96 of main text):

Figure S2. The niche is present in adult males and virgin females but not larvae.

- Dose response in adult male flies; *Ai* fed to germ-free flies (*Ai*->GF), *LpWF* fed to germ-free flies (*LpWF*->GF), *Ai* fed to flies colonized with *LpWF* (*Ai*->*LpWF*), *LpHS* fed to germ-free flies (*LpHS*->GF); error bars show standard error of the percentage (SEP).
- Steady state abundance in adult male flies (n=36 flies, >3 biological replicates).
- Max intensity Z-projection of *Ai* colonization in adult male proventriculus.
- Max intensity Z-projection of *LpWF* colonization in adult male proventriculus.

- E. Max intensity Z-projection of colonization *LpHS* in adult male proventriculus.
- F. Dose response in virgin female flies; error bars are SEP.
- G. Steady state abundance in virgin female flies (n=72, >3 biological replicates).
- H. Max intensity Z-projection of *Ai* colonization in adult virgin female proventriculus.
- I. Max intensity Z-projection of *LpWF* colonization in adult virgin female proventriculus.
- J. Max intensity Z-projection of *LpHS* colonization in adult virgin female proventriculus.
- K. Max intensity Z-projection of *Ai* colonization in larval proventriculus. Yellow line marks inner lumen.
- L. Max intensity Z-projection of *LpWF* colonization in larval proventriculus. Yellow line marks inner lumen.
- M. Max intensity Z-projection of *LpHS* colonization in larval proventriculus. Yellow line marks inner lumen. Scale bars are 20 μ m

-Beyond this, providing further mechanistical insight on the strain specificity of the colonization phenotype or hints on the mechanism underlying niche construction remain the outstanding questions that this referee would like see addressed (at least partly) in a high impact publication.

As the reviewer suggested, we investigated the molecular basis of niche construction. We sought to characterize the secreted layer that we observed by electron microscopy. Due to the potential similarity to mucus, we hypothesized that the secreted layer is glycan-rich. Using a panel of glycan-specific lectins, we were able to characterize the secreted substance as *N*-acetylglucosamine-rich, and not rich in other glycans, including chitin, sialic acid, or *N*-acetylgalactosamine, that are known to decorate extracellular matrices. We further show that *N*-acetylglucosamine-rich secreted material is produced by the fly and not simply a digested product of food or bacteria. Thus, in addition to providing a physical niche, the host also secretes a chemical substrate for bacterial growth. It is congruent with exciting work from larval flies (see Storelli et al 2018 Cell Metabolism), showing that larval excrement contains *N*-acetylglucosamine, which promotes *L. plantarum* growth. New Figures 5 and S10 detail these results:

From the Introduction (line 64-68):

While colonization of the fly gut has long been argued to be non-specifically regulated by host filtering mechanisms, including feeding preferences, immunity, and digestion, recent evidence suggests flies may also selectively acquire *Lactobacillus* and *Acetobacter* strains in the wild (Obadia et al., 2017; Pais et al., 2018) and provide them nutrition during the larval phase (Storelli et al., 2018).

In the Results (line 240-269):

Lectin staining reveals a glycan-rich matrix associated with the foregut niche

Mucus is heavily glycosylated, with various glycan subunits including *N*-acetylglucosamine, *N*-acetylgalactosamine, *N*-acetylneuraminic acid, mannose, glucose, fucose, and arabinose, depending on the type of mucus (Johansson et al., 2013). We hypothesized that the extracellular matrix (ECM) in the colonized zone of the

proventriculus inner lumen, as seen by thin section TEM (Fig. 4, S9), is glycan-rich. To test this hypothesis, we sectioned the proventriculus and stained with a panel of lectins that have specificity for glycans found in mucus (Methods). Wheat germ agglutinin (WGA), *Dolichus biflorus* agglutinin (DBA), and *Lens culinaris* agglutinin (LCA) stained fly cells in the proventriculus sections, while other lectins showed no consistent binding to the proventriculus (Fig. 5A). Examining the inner lumen of the proventriculus where the secreted layer appears by TEM (Fig 4H-M, S9), we observed staining by WGA (Fig. 5B). DBA and LCA staining was also present in the inner lumen but the locations did not correspond to the secreted layer seen by TEM (Fig. S10A,B). WGA binds *N*-acetylglucosamine and can also bind *N*-acetylneuraminic acid (Monsigny et al., 1980), the major sialic acid found in mammalian cells. *Sambucus nigra* agglutinin, which binds sialic acids, did not stain, consistent with published reports that sialic acid occurs only in fly embryos (Roth et al., 1992). Succinylated WGA (sWGA) binds only *N*-acetylglucosamine (Monsigny et al., 1980) and exhibited a staining pattern consistent with WGA and TEM images of the secreted layer, indicating that the inner lumen ECM is *N*-acetylglucosamine-rich (Fig. 5C). Chitin is a polymer of *N*-acetylglucosamine, and a permeable, chitinous cuticle is also present in the proventriculus (King, 1988). Calcofluor, which binds chitin, stained the cuticle, which can be clearly seen between the inner lumen epithelial cells and the secreted layer, indicating that the secreted layer is not chitin (Fig. 5B,C). TEM suggested germ-free flies also have a narrow secreted layer (Fig. 4H). We analyzed germ-free flies and found that this narrow layer stained with WGA, indicating that *N*-acetylglucosamine is a primary glycan in the secreted layer of germ-free flies (Fig. 5D, S10C). To determine whether the layer could be the product of digested chitin from yeast cell walls in the food, we sectioned newly eclosed virgin females before their first feed and found that staining of their proventriculus was consistent with germ-free flies (Fig. 5E, S10D,E). Because the proventriculus is formed from an imaginal disk during pupariation, our evidence indicates the secreted *N*-acetylglucosamine-rich ECM is clearly produced by the fly.

Lectin	Target	Inner lumen cells	Inner lumen	Secreted layer
WGA	N-acetylglucosamine, N-acetylneuraminic acid	+	+	+
S-WGA	N-acetylglucosamine	+	+	+
DBA	N-acetylgalactosamine	+	-	-
LCA	mannose, glucose	+	+/-	-

Figure 5. Lectin staining reveals a glycan rich matrix associated with the foregut niche. (A) Table of lectins that stained the proventriculus. (B-E) Lectin staining of proventriculus transverse sections for (B) WGA in colonized fly, (C) sWGA in colonized fly, (D) WGA in germ-free fly, and (E) WGA staining in newly eclosed germ-free fly before first food ingestion. Scale bars 20 μ m.

Figure S10. Lectin staining of the proventriculus.

- A. DBA staining a transverse cross section of a colonized proventriculus.
- B. LCA staining a transverse cross section of a colonized proventriculus.
- C. WGA staining a transverse cross section of a germ-free proventriculus.
- D. LCA staining a transverse cross section of a newly eclosed germ-free proventriculus.
- E. sWGA staining a transverse cross section of a newly eclosed germ-free proventriculus. Scale bars are 20 μ m.

Specific points:

-Previous publications are not referenced properly in the supplementary materials file and currently cannot be matched with the main document references.

We corrected the references.

-FigS1K would benefit from having similar quantification as seen in FigS1J.

We have added this quantification as Figure S1L. See below.

-line 130: I believe FigS2G should be referenced instead of FigS2F

Thank you for catching this. It has been corrected.

Reviewer #2 (Remarks to the Author):

The manuscript by Dodge et al. presents a good body of work characterizing a physical niche in the Drosophila gut occupied and shaped by efficient bacterial colonizers. The data are solid in showing the different types of ecological interactions between initial and secondary colonizers, analogous to the notions colonization resistance and facilitation established in the mammalian microbiome studies. The paper is well written and the knowledge is valuable to the microbiome field.

We appreciate the reviewer's positive appraisal of the work.

Some comments:

1) L31-32: The significance of the study on host genetics and discovery of new niches in animals is over-extended and lacks support in the main text.

We removed that line.

2) The proventriculus morphological change seems to be specifically dependent on

LpWF and happen as quick as 1hpi. The authors suggest this is due to host response to some unknown bacterial molecules, not simply physical pressure. The authors also point out a mucus-like zone between the intestine wall and the lumen upon LpWF colonization. The reviewer finds it curious and valuable to provide at least some insights into the host response that leads to the mucus-like zone. To see if the host response is dependent on LpWF cell wall or active metabolism, one simple experiment would be to examine luminal space expansion upon 1hpi of heat-killed LpWF. How chemically similar are the gut mucosa between Drosophila and animals? An RNAseq experiment shortly after LpWF introduction would gain further mechanistic insights into the host response to niche occupancy.

None of these questions is as simple to resolve as the reviewer suggests nor can they be addressed definitively with "one simple experiment." Such an experiment would likely raise another question and then another. Characterizing the molecular nature of the host-bacterial interactions that provide the observed specificity, whether it turns out to be the mucus layer or some target, is the logical next major goal of our research program. Bringing these questions to a satisfying level will likely require a complete new study to establish definitively how the niche operates on a molecular level. Even in broad outline, it is likely that only the first of the key molecules could be identified.

To begin to address the molecules involved, we used lectin staining, which is a classical approach to characterize the glycosylation of extracellular matrices. Based on the work of Storelli et al 2018 Cell Metabolism, where they found that *Drosophila* larvae excrete an *N*-acetylglucosamine-rich substance that benefits *L. plantarum*, we hypothesized that the adult proventriculus niche secretes *N*-acetylglucosamine into the region that *L. plantarum* occupies. We performed experiments that demonstrated the presence of *N*-acetylglucosamine in the furrows of the niche and surrounding the bacterial cells. Please see our response to Reviewer #1. To summarize, other common glycans found on mucus were either not found or were not localized to the layer of niche expansion. The *N*-acetylglucosamine-rich substance was also present in germ-free flies but to a lesser extent, corresponding to the secreted layer that is visible by TEM. We performed control experiments in germ-free, newly eclosed flies prior to their first meal that rule out staining of bacteria or food. This biochemical characterization of a key molecular component of the niche which promotes the growth of the niche's key occupant underscores the symbiotic relationship between the host and its symbiotic bacteria. These results appear as new figures, Figure 5 and Figure S10, lines 240-269 of the main text (and see response to Reviewer #1).

Reviewer #2 also suggested the heat-killed *LpWF* experiment as a way to further differentiate the trigger of the host response to niche occupancy between passive *LpWF* cell wall versus active *LpWF*. We performed this experiment. We found that heat-killed *LpWF* is not able to facilitate *Ai* colonization (New Fig. S7A-B; lines 203-4) or cause niche expansion (New Fig. S7C-F; lines 229-231).

Figure S7. Heat-Killed *Lactobacillus plantarum* (*LpWF*) does not elicit a host-response

- Schematic of priority effects experiment in heat-killed treated flies; flies were fed heat-killed *LpWF* as in (A), then fed doses of *Ai* bacteria.
- Dose response for *Ai* in germ-free and *Ai*->*LpWF* in heat-killed treated flies (n=72 flies).
- Schematic of heat-killed experiment; to mimic colonization by *LpWF*, 5-7do mated female flies were fed with heat-killed *LpWF* bacteria daily for 3 days, the gut was then cleared of excess killed bacteria by transferring to fresh food overnight and then to agar-water for 4 hours before embedding and freezing for cryosectioning.
- Cross section of anterior proventriculus in *LpWF*-colonized flies stained with calcofluor. Furrows are expanded by presence of bacteria. D'. Color rendering of D with bacteria shown. Yellow = *LpWF*-mCherry bacteria, blue=cuticle stained with calcofluor.
- Cross section of germ-free anterior proventriculus stained with calcofluor. Furrows are more narrow than for colonized.
- Heat-killed treated anterior proventriculus stained with calcofluor. Furrows are not significantly enlarged following 3 days feeding with *LpWF*. All scale bars are 10 μ m.

3) Grammar: L68 an physical niche  a physical niche; L180 co-localized with each other a furrow  co-localized with each other in a furrow

Thank you, we corrected these.

Reviewer #3 (Remarks to the Author):

Dodge et al report that a physical niche regulates stable association of a multispecies gut microbiota. The researchers use germ-free, and gnotobiotic Drosophila, which are techniques that have been widely used in many labs over the past decade, due to the comparative straightforward methodology of generating germ-free flies. This model has been widely used for the mono-association, or gnotobiotic colonization of a known cocktail of bacteria. One strength of the manuscript is the high-quality imaging of the Drosophila whole alimentary canal. These techniques have continued to develop since some of the initial detailed characterization of physiological compartmentalization within the Drosophila midgut (PMID: 23991285). Using bacterial strains that are tractable within the Drosophila intestine is also a strength. However, despite these powerful imaging methodology, the biological conclusions drawn from the experiments only represent modestly advancements to the field. Due to the reductionist nature of the model, with only a couple of bacteria tested, certainly the title of the manuscript is rather overreaching in its conclusion, especially extrapolating to infer “multispecies gut microbiota”. These interactions might be of Lactobacillus and Acetobacter might be highly specific to this model. Further examples of bacteria that physically remodels and intestinal niche to favor secondary colonization by an unrelated commensal would be required to make such generalizable conclusions.

We believe our findings represent a major advance, because previously no bacterial species has been shown to associate with a specific region of the *Drosophila* gut, with the characteristics of niche colonization as documented here. Spatial specificity in the gut is highly significant for understanding the physiological basis of microbiome-host interactions, because of the spatial compartmentalization within the midgut as the reviewer pointed out. The foregut niche we establish here provides a model for studying an exceptionally tightly defined spatial specificity in a major secretory tissue, the proventriculus. Certainly any model system will have specific attributes, but it is the nature of model systems that they allow mechanistic dissection of a broader phenomenon, namely spatial specificity between host and microbe in the gut. How an intimate relationship such as this one is maintained has generalizable features, such as the regulation of symbiotic secretions.

Regarding whether the results are generalizable to other bacterial species, the facilitation provided by *LpWF* is generalizable to other *Acetobacter* from a separate clade (Pitiwittayakul et al., 2015), as we show below for *A. pasteurianus* (New Fig. S6D; lines 201-203):

A. pasteurianus, a phylogenetically distinct species of *Acetobacter* (Pitiwittayakul et al., 2015) that is common in *D. melanogaster* (Sannino et al., 2018), was also facilitated by *LpWF* (Fig. S6D).

Here, *A. pasteurianus*, an *Acetobacter* commonly found in flies, is shown to colonize poorly in germ-free flies, both with a low percentage of flies colonized regardless of dose (left) and with a low abundance of only a few hundred cells in the flies that are colonized at all (right). However, when *LpWF* colonizes the flies first, *A. pasteurianus* colonizes with greater efficiency (left) at an abundance of ~1000 CFUs in colonized flies (right).

REVIEWER COMMENTS

Reviewer #1 (Remarks to the Author):

In this revised MS the authors provide additional information and new experimental results that address in a very satisfactory manner my previous comments.

Reviewer #3 (Remarks to the Author):

Dodge et al report that a symbiotic physical niche regulates stable association of a multispecies gut microbiota. They conclude that specific strains of fly gut bacteria colonize crypt-like furrows in the proventriculus, and that the proventriculus responds to colonization through engorgement, which subsequently promotes colonization by bacteria. The study is comprehensive and noted by cutting edge imaging methodology of the fly intestine. I have the following comments:

Is the LpWF heterofermentative or homofermentative. i.e. does the primary by-product that LP produces govern their ability to facilitate the colonization of secondary bacteria such as Ai to the niche?

It has been reported that *Lactobacillus rhamnosus* GG (LGG) requires the SpaC protein to attach to mucus. Is there anything known about a SpaC-like protein in LpWF ?

In line 235, the authors state, "This morphology is reminiscent of mammalian mucus, which has two layers: a dense, uncolonized layer adjacent to the epithelium, and a thinner, distal layer colonized by bacteria (Altmann, 1983)." Do flies have homologues of genes in mammals that function in mucus production? Is there a difference in the expression of these genes in the fly gut between GF and conventional, or GF and Mono-colonized flies with LpWF, LpWCFS1, or Ai?

Can the fly gut be stained with a Periodic acid-Schiff staining to detect mucus or mucus-like elements?

Does the proventriculus change morphology following mono colonization with the LpWCFS1 strain isolated from humans (the LP that does not colonize)?

In line 302, the discussion regarding the proventricular furrows/niche is a little thin. I was expecting some degree of speculation about what is special about cells within the proventricular furrows/niche that endows them to facilitate the phenotype.

REVIEWER COMMENTS

Reviewer #1 (Remarks to the Author):

In this revised MS the authors provide additional information and new experimental results that address in a very satisfactory manner my previous comments.

We thank the reviewer for their time and consideration.

Reviewer #3 (Remarks to the Author):

Dodge et al report that a symbiotic physical niche regulates stable association of a multispecies gut microbiota. They conclude that specific strains of fly gut bacteria colonize crypt-like furrows in the proventriculus, and that the proventriculus responds to colonization through engorgement, which subsequently promotes colonization by bacteria. The study is comprehensive and noted by cutting edge imaging methodology of the fly intestine. I have the following comments:

We thank the reviewer for their time and consideration. We also appreciate their comments that the study is comprehensive and makes use of cutting edge methodology.

Is the *LpWF* heterofermentative or homofermentative. i.e. does the primary by-product that *LP* produces govern their ability to facilitate the colonization of secondary bacteria such as *Ai* to the niche?

We appreciate the reviewer's suggestion that the facilitation of *Ai* colonization by *LpWF* may be metabolically driven. *L. plantarum* is widely reported to be homofermentative. In published work, several labs including our own have reported that *L. plantarum* spent medium supports growth of various *Acetobacter* species [1,2]. The Ludington and Huang labs showed that *LpWF* spent media supports the growth of the *Ai* strain used in the present work [3]. We agree with the reviewer that there is likely a metabolic component to the facilitation; exploring this metabolic component is clearly beyond the scope of the current study. We have added a sentence to the Discussion speculating about the metabolic basis of facilitation.

See lines 299-301:

Indeed, *L. plantarum* is homofermentative, and we have previously shown that *LpWF* has a positive metabolic interaction with *Acetobacter* strains found in the fly gut, including *Ai* (Aranda-Díaz et al., 2020).

It has been reported that *Lactobacillus rhamnosus* GG (LGG) requires the SpaC protein to attach to mucus. Is there anything known about a SpaC-like protein in *LpWF* ?

There is no published genome for *LpWF*, but over 300 *L. plantarum* genomes are published, and all have at least two mucin-binding motifs like the SpaC protein. Thus, there is likely a protein that could be performing a function like SpaC in *LpWF*; indeed, we expect that some surface receptor protein must be playing that role. Identifying that protein is well beyond the scope of the present work, which the reviewer has already referred to as comprehensive. While we agree with the reviewer that this is an important future direction, we feel that adding a speculative genomic analysis would detract from the focus of the present study and lessen its impact. Related to one of the reviewer's

other comments asking for speculation about mechanism, we added the following text to the Discussion.

See lines 289-292:

We speculate that the niche cells in the proventricular furrows either produce the adhesion substrate for the bacteria

In line 235, the authors state, “This morphology is reminiscent of mammalian mucus, which has two layers: a dense, uncolonized layer adjacent to the epithelium, and a thinner, distal layer colonized by bacteria (Altmann, 1983).” Do flies have homologues of genes in mammals that function in mucus production? Is there a difference in the expression of these genes in the fly gut between GF and conventional, or GF and Mono-colonized flies with *LpWF*, *LpWCFS1*, or *Ai*?

Flies have mucin genes [4,5], and these have been found to be expressed in the general region of the proventriculus [6]. However, the proventriculus is a complex tissue [7], and which cell types within this tissue are producing the mucins and the effect of these mucins on colonization are more difficult questions that would require a completely new set of experiments and years of work to do in a convincing manner. Thus, these questions are clearly beyond the scope of the present study, which identifies the niche. As all of the reviewers have stated, the present work is comprehensive and convincing. To avoid diluting the impact of our work with open-ended speculation, we have moved the sentence in line 235 to the discussion and modified it to soften the comparison to human mucus.

See lines 286-293:

Furthermore, the morphology of the extracellular matrix in the proventriculus is reminiscent of mammalian mucus, which has two layers: a dense, uncolonized layer adjacent to the epithelium, and a thinner, distal layer colonized by bacteria (Altmann, 1983). We speculate that the niche cells in the proventricular furrows either produce the adhesion substrate for the bacteria or else produce an extracellular matrix that sequesters the adhesion substrate from another source, such as the salivary gland.

Can the fly gut be stained with a Periodic acid-Schiff staining to detect mucus or mucus-like elements?

Periodic acid-Schiff is a non-specific carbohydrate stain that stains substances as disparate as cellulose and glycogen; it does not unambiguously identify mucus. In the current manuscript, we used WGA, which serves a similar function to PAS. We then presented comprehensive evidence that the niche is stained by several carbohydrate stains that have more specificity than WGA or Periodic acid-Schiff, allowing us to determine the range of glycans that are enriched in the niche. In particular, we showed that N-acetylglucosamine is a primary glycan in the ECM of the niche. Thus, we do not think that using a less specific stain would add to our study given the results already presented.

No text has been modified with respect to this comment.

Does the proventriculus change morphology following mono colonization with the *LpWCFS1* strain isolated from humans (the LP that does not colonize)?

We provided TEM evidence that non-colonizing *Lp* does not change the morphology of the niche in Figure 4I. To address the reviewer's question, we have clarified in the text and figure legend that non-colonizing *Lp* is part of the microbiome in the fly used in that figure panel.

See lines 225-228:

Similar morphology was observed in conventionally reared lab flies, which are associated with poor-colonizing strains of the same bacterial species, including *L. plantarum* (Obadia et al., 2017) (Fig. 4I).

See also Figure 4I Legend:

(I) conventionally-reared fly (only lab fly bacteria; no *LpWF*)

In line 302, the discussion regarding the proventricular furrows/niche is a little thin. I was expecting some degree of speculation about what is special about cells within the proventricular furrows/niche that endows them to facilitate the phenotype.

To address the reviewer's point, we have added speculation about the cells within the proventricular niche. We think they could be directly producing the ECM in the niche or perhaps recruiting that material from another source, such as the salivary gland. We note that there is a yet unidentified structure in these cells in TEM images that appears to be stacks of membrane like the Golgi or ER and thus could function in secretion.

See lines 284-296:

Though we have not investigated mucins here, flies have mucins (Huang et al., 2022; Syed et al., 2008), and three of these are expressed in the proventriculus (Buchon et al., 2013). Furthermore, the morphology of the extracellular matrix in the proventriculus is reminiscent of mammalian mucus, which has two layers: a dense, uncolonized layer adjacent to the epithelium, and a thinner, distal layer colonized by bacteria (Altmann, 1983). We speculate that the niche cells in the proventricular furrows either produce the adhesion substrate for the bacteria or else produce an extracellular matrix that sequesters the adhesion substrate from another source, such as the salivary gland. The apparent densely stacked membranes observed by TEM of the niche cells (Fig. 1J) suggests these may be secretory cells, which is consistent with the extensive secretory nature of the proventriculus (King, 1988).

- 1 Consuegra, J. *et al.* (2020) Metabolic cooperation among commensal bacteria supports *Drosophila* juvenile growth under nutritional stress. *iScience*
- 2 Henriques, S.F. *et al.* (2020) Metabolic cross-feeding in imbalanced diets allows gut microbes to improve reproduction and alter host behaviour. *Nat. Commun.* 11, 4236
- 3 Aranda-Díaz, A. *et al.* (2020) Bacterial interspecies interactions modulate pH-mediated antibiotic tolerance. *Elife* 9,
- 4 Syed, Z.A. *et al.* (2008) A potential role for *Drosophila* mucins in development and physiology. *PLoS One* 3, e3041
- 5 Huang, Y. *et al.* (2022) JiangShi: a widely distributed Mucin-like protein essential for *Drosophila* development. *bioRxiv*
- 6 Buchon, N. *et al.* (2013) Morphological and molecular characterization of adult midgut

- compartmentalization in *Drosophila*. *Cell Rep.* 3, 1725–1738
- 7 King, D.G. (1988) Cellular-Organization and Peritrophic Membrane Formation in the Cardia (Proventriculus) of *Drosophila-Melanogaster*. *J. Morphol.* 196, 253–282